

# Effects of passages through an insect or a plant on virulence and physiological properties of the fungus *Metarhizium robertsii*

Oksana G. Tomilova[1,2], Vadim Y. Kryukov[1], Natalia A. Kryukova[1], Khristina P. Tolokonnikova[1], Yuri S. Tokarev[2], Arina S. Rumiantseva[2], Alexander A. Alekseev[1,3] and Viktor V. Glupov[1]

[1] Institute of Systematics and Ecology of Animals SB RAS, Novosibirsk, Russia
[2] All-Russian Institute of Plant Protection, St. Petersburg, Russia
[3] Voevodsky Institute of Chemical Kinetics and Combustion SB RAS, Novosibirsk, Russia

Corresponding author
Oksana G. Tomilova,
toksina@mail.ru

## ABSTRACT

Species of the genus *Metarhizium* are characterized by a multitrophic lifestyle of being arthropod parasites, rhizosphere colonizers, endophytes, and saprophytes. The process of adaptation to various organisms and substrates may lead to specific physiological alterations that can be elucidated by passaging through different hosts. Changes in virulence and cultivation properties of entomopathogenic fungi subcultured on different media or passaged through a live insect host are well known. Nevertheless, comparative in-depth physiological studies on fungi after passaging through insect or plant organisms are scarce. Here, virulence, plant colonization, hydrolytic enzymatic activities, toxin production, and antimicrobial action were compared between stable (nondegenerative) parent strain *Metarhizium robertsii* MB-1 and its reisolates obtained after eight passages through *Galleria mellonella* larvae or *Solanum lycopersicum* or after subculturing on the Sabouraud medium.

The passaging through the insect caused similar physiological alterations relative to the plant-based passaging: elevation of destruxin A, B, and E production, a decrease in protease and lipase activities, and lowering of virulence toward *G. mellonella* and *Leptinotarsa decemlineata* as compared to the parent strain. The reisolates passaged through the insect or plant showed a slight trend toward increased tomato colonization and enhanced antagonistic action on tomato-associated bacterium *Bacillus pumilus* as compared to the parental strain. Meanwhile, the subculturing of MB-1 on the Sabouraud medium showed stability of the studied parameters, with minimal alterations relative to the parental strain. We propose that the fungal virulence factors are reprioritized during adaptation of *M. robertsii* to insects, plants, and media.

## INTRODUCTION

Entomopathogenic fungi of the genus *Metarhizium* are widespread in terrestrial ecosystems of the world and demonstrate multitrophic lifestyles. These fungi are primarily arthropod parasites, and therefore they are exploited as microbial control agents against herbivore and blood-sucking insects (*Jaronski & Mascarin, 2016*; *Islam et al., 2021*). In addition to parasitizing, the fungi exhibit rhizosphere competency and endophytic activity thereby promoting plant growth and immunity (*St Leger & Wang, 2020*). Being soil inhabitants, *Metarhizium* fungi are concentrated mainly in the rhizosphere and may penetrate plant tissues as well (reviewed by *Vega (2018)*). The ability of *Metarhizium* spp. to colonize roots, stems, and leaves of various plant species in the laboratory (*e.g.*, *Rios-Moreno et al., 2016a*; *Krell et al., 2017*) and under field conditions (*Clifton et al., 2018*; *Tomilova et al., 2020*) has been documented. The fungi are characterized by specific sets of physiological adaptations both for insect infection (adhesion to the cuticle followed by its hydrolysis, secondary-metabolite synthesis, and insect immune response avoidance) and for plant colonization (*e.g.*, specific-adhesin synthesis and a xylose-metabolizing ability underlying insect-derived–nitrogen transfer to a plant) (reviewed by *Hu & Bidochka (2019)* and *St Leger & Wang (2020)*). Genetic and physiological changes of *Metarhizium* and *Beauveria* fungi can occur during adaptation to various hosts and substrates (*Butt et al., 2006*; *Hu & Bidochka, 2020*); however, the understanding of evolutionary ecology of these fungi is still limited (*Raymond & Erdos, 2022*). Adaptive changes in the physiology of *Metarhizium* species can be analyzed on the basis of passaging through artificial media or insect and plant organisms.

The attention of researchers has been focused mainly on studying the effect of long-term subculturing of strains on media because this topic is extremely important for the production of mycoinsecticides. The phenomenon of degeneration caused by successive subculturing of entomopathogenic fungi is not uncommon and includes decreases in virulence and sporulation levels, appearance of sectors in colonies, lowering of enzymatic activities and of secondary-metabolite production, and other deteriorative changes (reviewed by *Butt et al. (2006)*). In contrast, other fungal strains are able to retain their stability through successive subculturing (*e.g.*, *Ansari & Butt, 2011*; *Eckard et al., 2014*; *Hussien et al., 2021*). *Butt et al. (2006)* believe that the main reasons for the phenotypic and physiological degeneration are a transposable element, dsRNA virus activities, DNA methylation, and chromosome polymorphism. The loss of virulence in *Metarhizium* has been found to be linked with a decline of protease Pr1 activity, of toxin production, and of conidial adhesion force (*Shah & Butt, 2005*; *Butt et al., 2006*; *Shah et al., 2007*). In a article by *Jirakkakul et al. (2018)*, successive subculturing of *Beauveria bassiana* was proved to induce potent oxidative stress. In a proteomic study on fungal conidia after serial passaging, proteins involved in the oxidative stress response, autophagy, and apoptosis were found to be upregulated, while those responsible for DNA repair, ribosome biogenesis, energy metabolism, and virulence were downregulated (*Jirakkakul et al., 2018*).

Fungal strain degeneration during subculturing is a reversible process. *In vivo* passaging is considered an effective way to restore the virulence and conidial yield of

entomopathogens (*Butt et al., 2006*). Many reports indicate virulence enhancement when fungi are passaged through arthropods (*e.g.*, *Shah, Wang & Butt, 2005*; *Adames et al., 2011*; *Safavi, 2012*; *Jirakkakul et al., 2018*; *Hu & Bidochka, 2020*). Given the current information about the substantial role of plants in the life of entomopathogenic fungi, it is increasingly relevant to study the effect of passages through plants on fungal physiology. Nevertheless, research on passaging of insect pathogenic fungi through a plant organism can be found in only a couple of articles. In particular, *Hu & Bidochka (2020)* have shown that five-fold passaging through plants (soldier bean and switchgrass) and an insect (wax moth) restores conidia production and virulence of a degenerative strain of *M. robertsii*; in that study, the respective changes at the molecular level included a decline of both DNA methyltransferase expression level and of the number of specific hypermethylated regions in DNA. This result indicates substantial involvement of epigenetic changes—mediated by DNA methylation—in the reported phenomena. *González-Mas et al. (2021)* showed that triplicate passaging of *B. bassiana* through melon, tomato, and cotton does not alter virulence toward *Galleria mellonella* but enhances endophytic colonization of the plants. Comparative studies on transformations of stable (nondegenerative) strains of entomopathogenic fungi during passaging through insects, plants, or artificial media have not been carried out previously. This research can help to control the stability of strains during subculturing and offers an opportunity for their improvement *via* passaging through live hosts. In addition, this approach can help predict the behavior of strains after their introduction into ecosystems and could be interesting from the standpoint of polyfunctional biocontrol of phytophages and phytopathogens.

Changes in activities of virulence factors under the influence of serial passaging through media or living organisms are important for understanding endophytic/parasitic lifestyle adaptions of fungi for their stability in culture. Such factors include primarily hydrolytic enzymes and secondary metabolites (*Schrank & Vainstein, 2010*). Among hydrolases as virulence factors, the most important are proteases, endochitinases, and lipases (*Mondal et al., 2016*; *Dhawan & Joshi, 2017*; *Gebremariam, Chekol & Assefa, 2022*; *Ferreira & de Freitas Soares, 2023*). Proteases metabolize cuticular proteins during penetration (*St Leger et al., 1995*; *St Leger et al., 1996a*; *Dhar & Kaur, 2010*). Endochitinases act directly on chitin, which is a major constituent of the insect cuticle (*Oyeleye & Normi, 2018*; *Goughenour et al., 2021*). Lipases are indispensable for assimilation of host's nutrients (primarily from the insect fat body) that ensure fungal viability and conidial production (*Keyhani, 2018*). Moreover, lipases may be important for fungal development on or in the cuticle to break cuticular lipids down by hydrolysis of ester bonds in lipoproteins, lipids, and waxes (*da Silva et al., 2010*; *Supakdamrongkul, Bhumiratana & Wiwat, 2010*; *Sánchez-Pérez et al., 2014*; *Liu et al., 2019*). For instance, inhibition of lipase activity in *Metarhizium* by ebelactone B reduces fungal virulence to the cattle tick *Rhipicephalus microplus* (*da Silva et al., 2010*).

Secondary metabolites may also be essential for development of fungi in their hosts. In *Metarhizium*, major exometabolites are destruxins, which are produced in fungi during their growth in insects, their cadavers, media, and plants (*Vey, Hoagland & Butt, 2001*; *de Bekker et al., 2013*; *Golo et al., 2014*; *Rios-Moreno et al., 2016a*). These biomolecules are

well known for their ability to impair calcium channels and the cytoskeleton in hemocytes and to activate apoptosis, thus suppressing phagocytosis and encapsulation, which are important in a host immune response to the fungal invasion (*Charnley, 2003*; *Lu & St Leger, 2016*). The destruxin profile and production vary among *Metarhizium* species having different host ranges (*Wang et al., 2012*) and between strains within one species (*Rios-Moreno et al., 2016b*), and their role in virulence is ambiguous (*Donzelli et al., 2012*). The functions of destruxins produced by fungi in plants is not yet clear. Probably, these metabolites participate in communication with host plants as modulators of the immune system (*Pedras et al., 2001*; *Barelli et al., 2022*). Some authors (*Golo et al., 2014*; *Rios-Moreno et al., 2016a*) have also hypothesized that these biomolecules facilitate plant protection from insect pests because these substances have antifeedant properties and oral and contact toxicity (*e.g.*, *Amiri, Ibrahim & Butt, 1999*; *Thomsen & Eilenberg, 2000*; *Lodesani et al., 2017*). It has been shown that destruxin production diminishes in degenerative strains of *Metarhizium* (*Wang, Skrobek & Butt, 2003*; *Shah & Butt, 2005*), though these processes have not been analyzed under conditions of passaging through different hosts.

Besides the interplay of entomopathogenic fungi with insects and plants, such fungi also interact with other microorganisms inhabiting various niches, be that living organism tissues or an external environment. Entomopathogenic fungi can inhibit and can be inhibited by soil- and host-associated microorganisms (*Jaronski, 2007*; *Toledo et al., 2015*; *Lozano-Tovar et al., 2017*; *Boucias et al., 2018*; *Chertkova et al., 2023*). Antimicrobial properties of fungi can be mediated by toxins (*Fan et al., 2017*), volatile organic compounds (*Hummadi et al., 2022*), or antimicrobial peptides (*Tong et al., 2020*). To our knowledge, however, there is no information on how the antagonistic activity of entomopathogenic fungi is modified by passaging through different hosts.

The aim of this work was comparative analysis of physiological alterations in a stable (nondegenerative) strain of *M. robertsii* (MB-1) after passaging through an insect, plant, or medium. We wanted to determine how development in different hosts alters properties of the fungal entomopathogen, *e.g.*, morphological and culture characteristics, levels of plant colonization, virulence to insects, hydrolytic enzymatic activity, destruxin production, and antagonism toward bacterial and fungal phytopathogens.

## MATERIALS AND METHODS

### Fungi, insects, and plants

A culture of the entomopathogenic fungus *Metarhizium robertsii* J.F. Bisch., S.A. Rehner & Humber (Hypocreales, Clavicipitaceae) from the Collection of Microorganisms at the Institute of Systematics and Ecology of Animals, the Siberian Branch of the Russian Academy of Science (ISEA SB RAS) was used. Parental strain MB-1 (GenBank accession # OR060959) was originally isolated from soil in a forest-steppe zone of Novosibirsk Oblast (Western Siberia) in 2009. A partial sequence of translation elongation factor (5′EF-1a) was employed to determine the species identity (*Kryukov et al., 2017*). The culture was maintained on ¼ Sabouraud Dextrose Agar with 0.25% of yeast extract (SDAY) at 4 °C and subcultured on the annual basis.
A laboratory culture of the greater wax moth *Galleria mellonella* L. (Lepidoptera: Pyralidae) from the Siberian population was used. Morphological and immunological characteristics, diet, and life history of the line are described in detail by *Dubovskiy et al. (2013)*. Sixth-instar larvae were used in all assays. Fourth-instar larvae of the Colorado potato beetle *Leptinotarsa decemlineata* Say (Coleoptera: Chrysomelidae) were collected in private potato fields in Novosibirsk Oblast (53°44′03″N, 77°39′00″E). Tomato *Solanum lycopersicum* L. (Solanales: Solanaceae) plants of the variety Beliy naliv 241 (SeDeK, Moscow, Russia) were employed in the experiments.

## Fungal-strain passaging

### Overall design

Prior to the experiments, conidia samples of the fungus were either plated on SDAY supplemented with lactic acid (0.2%) for passaging experiments or stored in an aqueous glycerol solution at −80 °C. An independent series of eight passages was performed (a) on SDAY with lactic acid, (b) in the wax moth larvae, or (c) in the tomato plants. After each passage though the insect or plant, the reisolates were plated on SDAY with lactic acid to standardize conidia production (which cannot otherwise be ensured in plants). The parent culture was stored at −80 °C until the last the passage. After the eighth passage, respective samples from the SDAY, plant, insect, and parental strain were plated on the medium and subjected to morphological examination, culture characterization, virulence tests, and antagonistic and biochemical assays. Before the assays, all cultures were checked for viability of conidia. The germination rate on SDAY was >95%.

### Passaging through the insect

Inoculation of insects with the fungi and reisolation of cultures were conducted by standard techniques (*Eckard et al., 2014*; *Hussien et al., 2021*) with a minor modification. The sixth-instar larvae of *G. mellonella* were inoculated by 15 s immersion into a $10^8$ conidia/mL suspension in water supplemented with 0.03% of Tween 80. Control insects were treated with an aqueous Tween 80 solution. The larvae were kept on an artificial diet in 90 mm Petri dishes at 26–27 °C, 90–95% relative humidity (RH), under a photoperiod of 0:24 h (L:D). Mummified cadavers (killed by a fungus at 6–7 days after the inoculation) were transferred to moisture chambers for conidiation (Fig. S1A) and were plated onto SDAY with lactic acid for further manipulations.

### Passaging through the plant

A modified approach of *Hu & Bidochka (2020)* was applied to the passaging of fungi through plants. Tomato seeds were sterilized with 0.5% sodium hypochlorite for 2 min and with 70% ethanol for another 2 min, followed by a triple wash in distilled water (*Posada et al., 2007*). Then, the seeds were inoculated by immersion in the aqueous Tween 80 (0.03%) suspension ($10^8$ conidia/mL) for 30 min (*Ahmad et al., 2020*) and sown onto sterile moistened sand (25 mL of sterile water per 150 g of sand) in 500 mL plastic containers. The containers were covered with perforated lids and kept in a climatic chamber at 24 °C and 75% RH with a photoperiod of 16:8 h (L:D). Control seeds were treated similarly but without the addition of the fungal conidia. Reisolates were obtained

from above-ground parts of plants 20 days after the inoculation. Surface-sterilized leaves and stems were plated on SDAY with lactic acid to ensure conidia production (Fig. S1B) for further manipulations.

### Passaging through the medium

Successive subculturing was conducted according *Ansari & Butt (2011)*. The parental strain was plated on SDAY with lactic acid and subcultured synchronously with the rounds of passaging through the insects and the plants at 26 °C in darkness.

## Colony morphological analysis

Standard media were used to grow fungal colonies: SDAY, potato-dextrose agar (PDA), and the minimal medium (MM: 6 g/L NaNO$_3$, 0.52 g/L KCl, 0.52 g/L MgSO$_4$7H$_2$O, 0.25 g/L KH$_2$PO$_4$). The MM was supplemented with 1% of xylose or glucose (*Xiao et al., 2012*). Morphological characteristics, including the color, shape, radial growth, and sector formation, were examined within 20 days after the plating. Four replicates were analyzed for each reisolate.

## Sequencing of an α/β-hydrolase gene fragment

To make sure that reisolates were successfully obtained by the subculturing and passaging as described above, genotyping was necessary. Given that multiple isolates of *Metarhizium* of different origins share an identical signature of the TEF sequence (*Kryukov et al., 2017*), there was a need to use a more variable locus. Accordingly, a protein belonging to the superfamily of α/β-hydrolases was chosen, previously exploited for differentiation between *Beauveria* strains (*Levchenko et al., 2020*). A set of sequences from GenBank, including XM007818060 (accession #) from *M. robertsii* and CP058937 from *M. brunneum*, were aligned in BioEdit (*Hall, 1999*). Primers Metaslip55F (5′-CTCCATAAAGAACATGTGTCCGTTGC-3′) and Metaslip1024R (5′-GGCAAATCTACGTCGAGAAGC-3′) were selected manually and then checked for compatibility in PerlPrimer (*Marshall, 2004*). To investigate variation of the locus chosen in *M. robertsii*, 10 strains from Novosibirsk Oblast were studied (from the Collection of Microorganisms at the ISEA SB RAS). For genomic DNA extraction, we utilized a simplified protocol of *Sambrook, Fritsch & Maniatis (1989)* without the addition of phenol. Standard PCR (*Malysh et al., 2020*) was run using the DreamTaq Green PCR Master Mix (Thermo Fisher Scientific, Waltham, MA, USA) on a Tertsik thermal cycler (DNK-Tekhnologiya, Moscow, Russia). Amplicons of expected size were separated by agarose gel electrophoresis, purified by DNA adsorption on silica particles (*Vogelstein & Gillespie, 1979*), and sequenced in both directions by the chain termination method (*Sanger, Nicklen & Coulson, 1977*) using an ABI Prism sequencer (Evrogen, Moscow, Russia). The obtained reads, 885 bp long, were aligned and visually checked for accuracy of automatic peak interpretation in the original chromatogram in BioEdit and were compared to the GenBank entries by means of BLAST.

## A virulence assay

The virulence of *M. robertsii* reisolates was assayed using either *G. mellonella* or *L. decemlineata*. Sixth-instar larvae of *G. mellonella* were immersed in a fungal conidia suspension ($10^8$ conidia/mL) and maintained as described above (see subsection "Passaging through the insect"). Mortality was scored daily for 13 days (until pupation). Seven replicates (one replicate = 15 larvae) for each reisolate were subjected to the assay. Control insects were treated with a conidia-free aqueous Tween 80 (0.03%) solution.

In the bioassay on *L. decemlineata*, the concentration of conidia was decreased to $10^7$ conidia/mL because this species is more susceptible to *M. robertsii* than *G. mellonella* is. Fourth-instar larvae of *L. decemlineata* were immersed into an aqueous Tween 80 (0.03%) suspension for 15 s and then transferred to fresh potato leaves in 300 mL ventilated plastic containers and maintained at 26 °C, 30–40% RH, and a photoperiod of 16:8 h (L:D). Feed was refreshed and mortality was scored daily for 8 days. Thirteen replicates (one replicate = 10 larvae) were assayed for each reisolate. Control insects were treated with an aqueous Tween 80 (0.03%) solution without the addition of the fungal conidia.

## A plant colonization assay

The plant seed treatment was performed as described above (see subsection "Passaging through the plant"). After 20 days of maintenance, upper parts of tomato seedlings were sterilized with hypochlorite and ethanol (*Posada et al., 2007*), imprinted (*McKinnon et al., 2017*), and plated in 90 mm Petri dishes (one plant per dish) containing a modified Sabouraud medium (glucose, 40 g/L; peptone, 10 g/L; yeast extract, 1 g/L; agar, 20 g/L) supplemented with cetyltrimethylammonium bromide (0.35 g/L), cycloheximide (0.05 g/L), tetracycline (0.05 g/L), and streptomycin (0.6 g/L) to inhibit the growth of saprophytic fungi and bacteria. The plates were incubated at 24 °C for 10 days, and plants colonized by *M. robertsii* were counted. Samples in which fungal growth was registered on prints were excluded from the analysis. There were 100 plants in each treatment group.

## A hydrolase assay

### Sample preparation

For this purpose, a modified method of *Ment et al. (2020)* was used. Submerged cultures were grown in MM broth with the addition of 1.2 g of the dry cuticle of *G. mellonella* per 800 mL of broth. For cuticle preparation, sixth-instar larvae were dissected, and their internal organs were removed by means of a spatula. The remaining cuticle was washed several times with saline (0.9% NaCl), lyophilized at −65 °C and 600 mTorr for 24 h, ground with a mortar and pestle in liquid nitrogen, and added to MM. The medium was poured into 50 mL conical flasks, autoclaved, inoculated with the fungal conidia grown for 14 days on SDAY at the concentration of $5 \times 10^6$ conidia/mL, and incubated for 8 days at 26 °C with constant shaking at 150 rpm. The fungal mycelium was sedimented by centrifugation at $5,000 \times g$ and 4 °C for 30 min, washed with Tris-buffered saline, and sonicated in an ice bath with three 10 s bursts on a Bandelin Sonopuls sonicator (GmbH & Co. KG, Berlin, Germany). The resulting suspension was centrifuged at $10,000 \times g$ and 4 °C

for 20 min, and a 1 mL sample of the supernatant was taken from each flask. Four biological replicates (flasks) of each culture were used for determining enzymatic activities.

### An assay of nonspecific protease activity

Proteolytic activity was measured with azocasein (Sigma-Aldrich, St. Louis, MO, USA) as a substrate by a method of *Zanphorlin et al. (2011)*, with modifications. A total of 40 μL aliquot of the supernatant (see subsection "Sample preparation") was added to 250 μL of 0.5% azocasein and 250 μL of 0.5 mM Tris-HCl buffer (pH 8.0) containing 0.15 mM NaCl. After incubation for 90 min at 28 °C, the reaction was terminated by the addition of 250 μL of 20% trichloroacetic acid. Next, the samples were cooled for 10 min at 4 °C and centrifuged at $20,000 \times g$ and 4 °C for 5 min. The enzymatic activity was determined spectrophotometrically on a 96-well plate reader at a wavelength of 366 nm.

### An assay of nonspecific lipase activity

The lipolytic activity was measured according to *Albro et al. (1985)* with minor modifications. A total of 40 μL aliquot of the supernatant (see subsection "Sample preparation") was added to 200 μL of 24 mM $NH_4HCO_3$ (pH 8.5) containing 0.4 mM *p*-nitrophenyl myristate and 8 mM Triton X-100. The enzymatic activity was determined spectrophotometrically on a 96-well plate reader at 405 nm.

All enzymatic activities were measured as a change of optical density units (ΔA) in the incubation mixture per minute per milligram of protein. The concentration of protein in the supernatant was determined by the Bradford method (*Bradford, 1976*). For construction of the calibration curve, bovine serum albumin was employed as a standard.

## Assessment of destruxin production

Quantification of destruxins (dtx) A, B, and E in culture broth was performed according to the method of *Seger et al. (2004)* with minor modifications. Conidia were added into conical flasks containing 25 mL of Czapek–Dox broth to obtain final concentration $5 \times 10^6$ conidia per milliliter of broth and were then incubated for 8 days at 26 °C and 150 rpm. Fungal biomass was removed by centrifugation ($20,000 \times g$, 30 min), and the pellets were dried at 70 °C to a constant weight. The supernatants were passed through a 0.22 μm nylon membrane filter (Millipore, Burlington, MA, USA). Aliquots of the resulting filtrates were diluted 1:1 with the acetonitrile for high-performance liquid chromatography (HPLC) with a diode array detector. An Agilent 1,260 Infinity HPLC system (Agilent Technologies, Santa Clara, CA, USA) was used equipped with a C18 column (Diaspher 110-C18, 2.1 mm × 150 mm, 5 μm particle size, BioChemMak ST JSC, Moscow, Russia). HPLC conditions were as follows: 30 °C column temperature and 0.4 mL/min flow rate, with recording of chromatograms at 210 nm. The injection volume was 5 μL. The mobile phase consisted of water (solution A) and acetonitrile (solution B). The following gradient was implemented: 0 min, 30% B; min 20, 50% B; min 21, 80% B; min 21–27, 80% B; and min 28–40, equilibration at 30% B. A calibration curve for dtxA (99% purity; Sigma-Aldrich, Saint Louis, MO, USA) was built from six concentrations (0.125, 0.25, 0.5, 1, 10, and 50 μg/mL) and was linear in this range ($R^2 = 0.999$). Because dtxA, dtxB, and dtxE are major compounds in HPLC analysis of *M. robertsii* culture broth and have identical sequences of

peaks during elution from a C18 column (*Seger et al., 2004*; *Wang et al., 2012*; *Golo et al., 2014*), despite the absence of dtxB and dtxE standards, we identified those compounds on the basis of UV spectra (Fig. S2) and literature data. Four biological replicates (flasks) of each reisolate were subjected to the experiment.

## An assay of antagonistic activity

### Antagonism toward the fungal phytopathogens

The antagonistic action of *M. robertsii* reisolates was assayed against fungal pathogens of plants from the Collection of Microorganisms at the All-Russian Institute of Plant Protection. Fungal cultures of *Fusarium oxysporum*, *F. solani*, *Rhizoctonia solani*, *Bipolaris sorokiniana*, and *Botrytis cinerea* were used in a cocultivation assay performed as described elsewhere (*Sobowale et al., 2010*), with modifications. An agar plug, 10 mm in diameter and containing the 5-day-old culture of *M. robertsii*, was placed in a 90 mm Petri dish filled with PDA at a distance of 3 cm from the dish margin. Two days later, a similar plug with a phytopathogen culture was placed in the same dish at a distance of 3 cm from the opposite margin of the dish. A phytopathogen culture grown in the absence of *M. robertsii* served as a control. The dishes were incubated at 26 °C in the dark. Phytopathogen growth inhibition by *M. robertsii* was estimated in comparison to radial growth of the control phytopathogen culture after 20 days of cultivation by means of a standard formula: $[(R1 − R2)/R1] × 100$ (*Barra-Bucarei et al., 2020*), where $R1$ is the radius (mm) of the control phytopathogen colony, and $R2$ is the radius (mm) of the pathogen colony competing against *M. robertsii*. Three replicates of each MB-1 reisolate were analyzed.

Additionally, in the case of *R. solani*, the density of mycelial primordia of the sclerotia was analyzed in the assay as described above. Inhibition of sclerotia formation was assayed by quantitation of mycelial color intensity in the ImageJ software (*Abramoff, Magelhaes & Ram, 2004*) in six replicates for each reisolate.

### Antagonism toward bacteria

The antagonistic activity of *M. robertsii* reisolates was assayed by the standard agar diffusion method against *Enterococcus faecalis* and *Enterobacter* sp., which are predominant in the *G. mellonella* gut (*Polenogova et al., 2019*), as well as against *Bacillus pumilus* isolated from the tomato plant seedlings. The bacterial species were identified by 16S ribosomal-RNA gene sequencing (*Polenogova et al., 2019*). A total of 10 mm agar plug with the 5-day-old *M. robertsii* culture was placed in the center of a 90 mm Petri dish containing PDA and freshly seeded with a bacterial culture. The dishes were incubated at 26 °C in darkness. The diameter of the sterile zone was measured on the 4th and 8th day in six replicates for each MB-1 reisolate.

## Statistics

The normality of data distribution was analyzed by the Shapiro–Wilk $W$ test. For normally distributed data, one-way ANOVA was applied followed by Fisher's least significant difference (LSD) test. For non-normally distributed data, Kruskal–Wallis ANOVA was applied followed by Dunn's test. The logrank test was performed to determine differences

in the mortality dynamics. Fisher's exact test was employed to compare plant colonization frequencies. Correlations between enzymatic activities, destruxin production, and virulence were determined by Pearson's correlation analysis. In graphs, the data are presented as the mean and standard error.

## RESULTS

### Morphology and genetics of the reisolates

#### Colony morphology and the rate of growth

When grown on four different media, the fungal colonies showed no sectorization. The morphology was very similar among the colonies obtained on different media, although on SDAY, the aerial mycelium was more prominent in all the passaged reisolates as compared to the parent strain (Fig. 1A). Radial growth rates were also similar, though in some cases, there were significant differences. In particular, on MM + xylose, the isolate passaged through the insect showed slower growth than did the parent strain and the plant-passaged culture (Fisher's LSD test, $p \leq 0.02$). On the contrary, the latter showed statistically significant growth retardation on MM + glucose ($p \leq 0.004$) as compared to the parent strain and the insect-passaged reisolate. The reisolate subcultured on SDAY demonstrated the fastest growth on this medium ($p \leq 0.039$) relative to the parent strain and the plant-passaged reisolate (Fig. 1B).

#### Molecular genetic analysis using the α/ß-hydrolase gene sequence

Ten strains of *M. robertsii* originating from Novosibirsk Oblast were found to belong to four haplotypes corresponding to separate lineages within the molecular phylogenetic tree of this species (Fig. 1C). Nucleotide sequence similarity between these haplotypes varied between 95.8% and 98.1%. The first, most prevalent haplotype was detected in five strains, including Q11 (GenBank accession # OP985336). No 100% matches for this sequence were found in GenBank. The second haplotype was presented by two strains from our collection, including 194-1 (# OP985337), as well as *M. robertsii* strain ARSEF 23, for which only an mRNA record is available (# XM007818060), and thus the intron sequence is not available. The third haplotype was detected in two strains from our collection, including MB-1 (# OP985335), without 100% GenBank matches. Finally, the fourth haplotype was assigned to a single strain: Q4 (# OP985338).

It should be noted that all the reisolates either subcultured on a medium or passaged through the plant or insect were 100% identical to each other and to parent strain MB-1. Meanwhile, only one of the remaining nine strains from our collection (sampled in Novosibirsk Oblast) was found to belong to the same haplotype. Therefore, MB-1 has the α/β-hydrolase molecular haplotype that is not very common, and we can presume that the reisolates obtained during the study truly originate from the parent strain because the probability of culture contamination with other samples of the same haplotype is negligible. The absence of *Metarhizium* cultures in samples from control (untreated) plants and insects confirms this assumption.

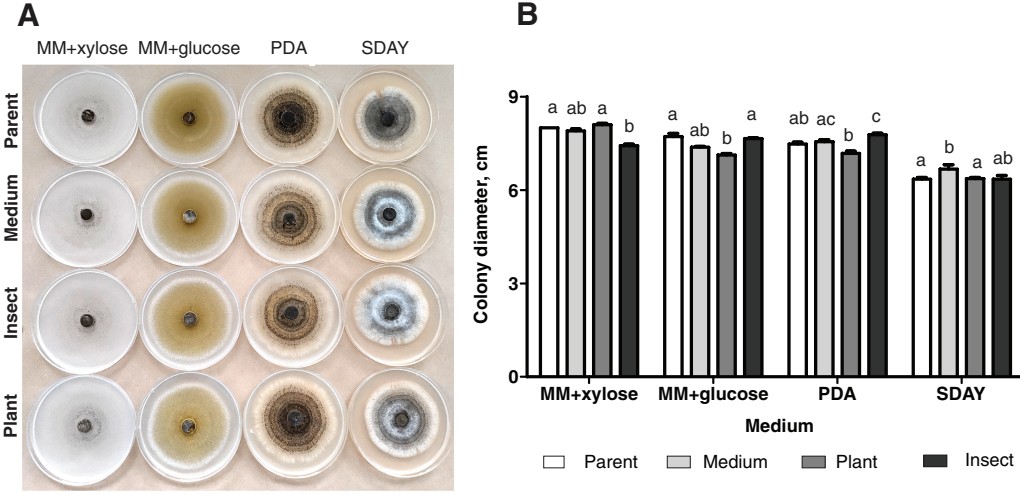

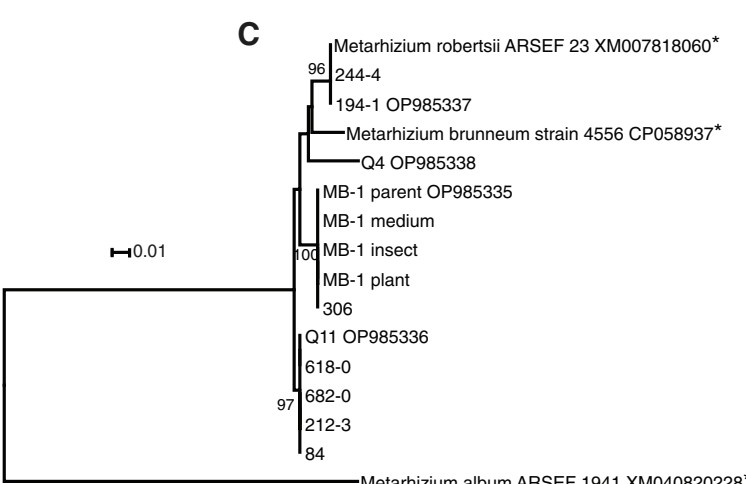

**Figure 1 Morphology and genotyping of *Metarhizium robertsii* MB-1 (parent) and its reisolates after eight cycles of subculturing on SDAY medium or after passaging through tomato or wax moth.** (A) Phenotypes of fungal colonies on different media 20 days after plating on minimal media (MM) with glucose or xylose, PDA, and SDAY. (B) Fungal colony diameter 15 days after plating on these media. Different letters indicate the significantly different values ($p < 0.05$, Dunn's test). (C) Phylogenetic positions of MB-1 reisolates and other *M. robertsii* strains (ISEA collection) as inferred from a maximum likelihood (ML) analysis based on the Tamura 3-parameter model (*Tamura & Nei, 1993*) of an alignment of partial α/β-hydrolase sequences, 878 bp long. The sequences downloaded from GenBank are pointed out by asterisks. ML bootstrap values are shown next to the branches. The tree is drawn to scale, with branch lengths measured as the number of substitutions per site (*Kumar, Stecher & Tamura, 2015*).

## Virulence levels

The mortality of the wax moth larvae caused by the fungal treatment did not exceed 60% on the 13th day post inoculation (d.p.i.) (Fig. 2A), and the highest score was observed in the case of the parent strain. The mortality dynamics induced by the reisolate subcultured on SDAY were not different from those of the parent strain (logrank test, $\chi^2 = 0.63$, df = 1, $p = 0.428$). Meanwhile, the speed of death and total mortality of the wax moth larvae under

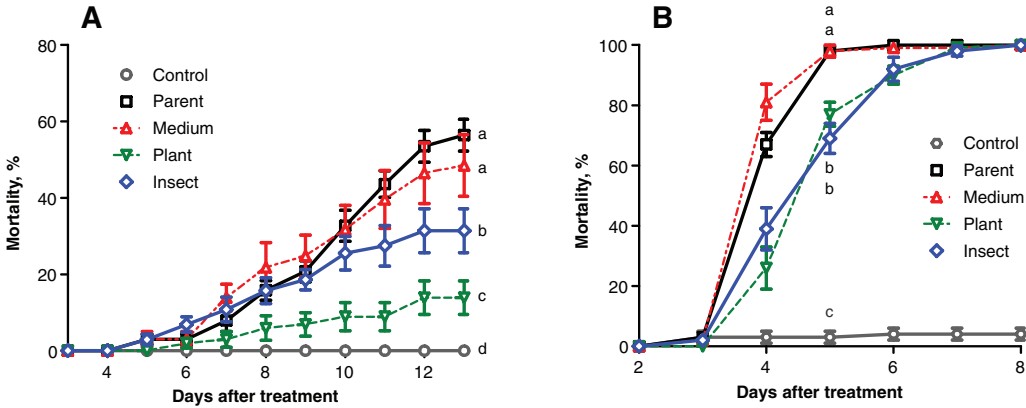

**Figure 2 Mortality dynamics of the test insects after inoculation with *Metarhizium robertsii* parent strain MB-1 and its reisolates from the SDAY medium, tomato and wax moth.** (A) Mortality of sixth instar larvae of wax moth, *Galleria mellonella* (*n* = 100 insects per treatment). (B) Mortality of fourth instar larvae of *Leptinotarsa decemlineata* (*n* = 130 insects per treatment). Different letters indicate significant differences (logrank test: $\chi^2 \geq 5.16$, df = 1, $p \leq 0.025$).

the influence of the plant-passaged and insect-passaged reisolates were significantly lower in comparison with the parent strain ($\chi^2 \geq 9.74$, df = 1, $p \leq 0.007$) and the SDAY-subcultured reisolate ($\chi^2 \geq 5.16$, df = 1, $p \leq 0.025$). Moreover, the plant-passaged reisolate was less virulent than the insect ($\chi^2 = 9.36$, df = 1, $p = 0.007$).

Similar results were obtained in the Colorado potato beetle larvae, though the mortality scores were higher, reaching 100% on the 5th–7th d.p.i. (Fig. 2B). Reisolates passaged through the insect and plant displayed a significant increase in median lethal time ($LT_{50}$) equaling 1 day when compared to the parent strain ($\chi^2 \geq 38.82$, df = 1, $p < 0.0001$) and the SDAY-subcultured reisolate ($\chi^2 \geq 52.19$, df = 1, $p < 0.0001$). The mortality dynamics did not differ between the infections by the plant and insect-passaged reisolates ($\chi^2 = 0.10$, df = 1, $p = 0.754$). As for the SDAY-subcultured reisolate, its virulence did not differ ($\chi^2 = 3.35$, df = 1, $p = 0.067$) from that of the parent strain either.

## Plant colonization

Frequencies of tomato seedings' colonization 20 days after the seed treatment were not significantly different between the reisolates and the parent strain. Colonization levels of tomato stems and leaves ranged between 18% and 28% (Fig. 3). Nonetheless, the plant colonization frequency tended to be higher (1.5-fold) in the plant-passaged reisolate compared to the parent strain and SDAY-subcultured reisolate with marginal significance (Fisher's exact test, $p = 0.091$ and $0.065$, respectively). In the insect-passaged reisolate, there was a similar trend without statistical significance ($p > 0.11$).

## Enzymatic activities

The lipase and protease activities tended to decrease in the passaged reisolates as compared to the parent strain (Fig. 4). When subcultured on SDAY or passaged through the plant or through insect, the fungal culture showed lower lipolytic activity as compared to the parent

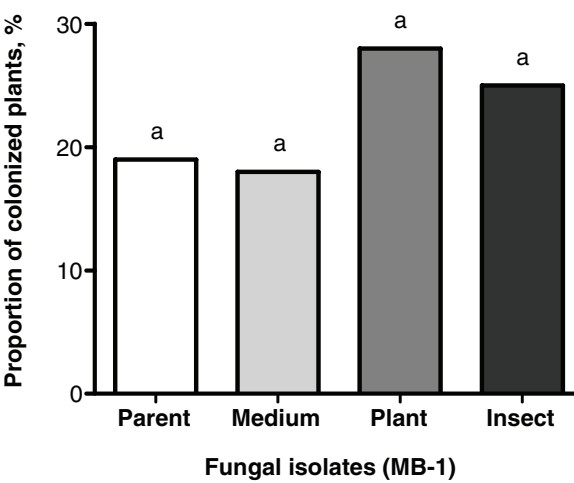

**Figure 3 Level of tomato seedling colonization by *Metarhizium robertsii* parent strain MB-1 and its reisolates (SDAY medium, tomato, and wax moth) on day 20 after inoculation.** Concentration of suspension used for inoculation was $10^8$ conidia/mL. Identical letters indicate non-significant differences (Fisher's exact test, $p \geq 0.065$; $n = 100$ plants per treatment group).

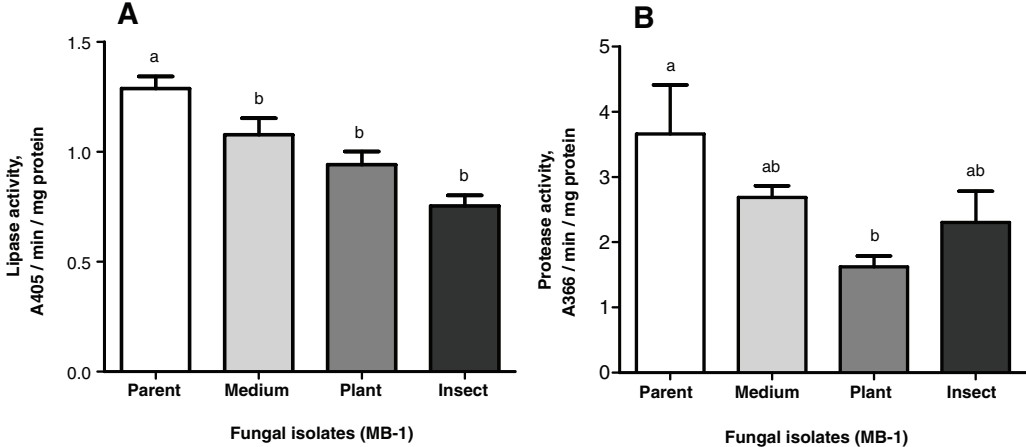

**Figure 4 Nonspecific enzyme activities of *Metarhizium robertsii* parent strain MB-1 and its reisolates (SDAY medium, tomato and wax moth).** (A) Lipase activity. (B) Protease activity. The fungus was cultivated in minimal medium supplemented by 1.5% of *Galleria mellonella* cuticles for 8 days ($n = 4$ per culture). Different letters indicate significantly different values (Fisher's LSD test, $p < 0.05$).

strain (1.2-fold, $p = 0.026$; 1.4-fold, $p = 0.014$; 1.7-fold, $p = 0.002$, respectively, Fig. 4A). There were no significant differences between the passaged cultures.

As for the proteolytic activity, it was 1.4- to 2.3-fold lower than that in the parent strain, though a statistically significant difference was observed only for the plant-passaged reisolate (Fisher's LSD test, $p = 0.010$); for the insect-passaged reisolate, the significance of the difference was marginal ($p = 0.065$) (Fig. 4B).

Notably, there was a statistically significant correlation between the proteolytic activity of *M. robertsii* isolates and their virulence to *G. mellonella*, as estimated *via* insect mortality

on the 13th d.p.i. (r = 0.951, $p$ = 0.049). Regarding the virulence to *L. decemlineata* (mortality on the 6th d.p.i.), the correlation was also positive but not significant (r = 0.877; $p$ = 0.123). Remarkable coefficients of correlation (r = 0.844–0.943) between lipolytic activity and virulence were seen only in the case of *L. decemlineata* (mortality on the 5th–7th d.p.i.), though statistical significance was marginal ($p \geq$ 0.057).

## Production of destruxins

Production levels of these toxins in the insect- and the plant-passaged reisolates proved to be elevated 1.8–3.1-fold as compared to the parent strain (Fisher's LSD test, dtxA: $p \leq$ 0.0002; dtxB: $p \leq$ 0.024; dtxE: $p \leq$ 0.0002, Fig. 5) and by 4.4–4.8-fold as compared to the SDAY-subcultured reisolate (dtxA: $p \leq$ 0.0002; dtxB: $p \leq$ 0.0005; dtxE: $p \leq$ 0.0002). In the latter, levels of the toxin production tended to be lower for all three destruxins assayed when compared to the parent strain, and in the case of destruxin A, the difference was significant ($p$ = 0.037).

A statistically significant negative correlation was observed between the virulence to *L. decemlineata* (4th–6th d.p.i.) and levels of production of all three destruxins (dtxA: r $\leq$ -0.951, $p \leq$ 0.049; dtxB: r $\leq$ -0.961, $p \leq$ 0.039; dtxE: r $\leq$ -0.984, $p \leq$ 0.016). There was a strong negative correlation between the virulence to *G. mellonella* (8th–13th d.p.i.) and destruxin production levels, although statistical significance was marginal (r between −0.915 and −0.756, $p \geq$ 0.085).

## Antagonistic activity toward phytopathogens

### Antagonism toward other fungi

The passaged reisolates showed minor, albeit in some cases statistically significant, changes in antimicrobial action on the phytopathogenic fungi (Fig. 6). The inhibitory activity of the insect-passaged reisolate against slowly growing *B. sorokiniana* was significantly higher relative to the parent strain (Dunn's test, $p$ = 0.02). The passaging through the plant enhanced inhibitory activity against fast-growing cultures of *R. solani* when compared to the parent strain ($p$ = 0.016) and against *B. cinerea* when compared to the SDAY-subcultured reisolate ($p$ = 0.016). Significant inhibition of sclerotia formation in *R. solani* was recorded under the influence of all the tested cultures, including the parent strain ($p \leq$ 0.0005, Fig. S3). Nonetheless, only the plant-passaged reisolate displayed an activity that was higher than that of the parent strain (Fisher's LSD test, $p$ = 0.021).

### Antagonism toward bacteria

No changes were detected between the parent strain and its reisolates in their antagonistic properties against the bacterium *E. faecalis* from the wax moth gut (Dunn's test, $p \geq$ 0.729, Fig. S4). As for *Enterobacter* sp., the antagonistic activity was weaker in the insect-passaged reisolate than in the parent strain and the plant-passaged reisolate ($p \leq$ 0.049, Fig. S4). Meanwhile, in the case of *B. pumilus* from the tomato seedlings, the bacterial growth was suppressed more strongly by the plant- and insect-passaged reisolates than by the SDAY-subcultured reisolate and parent strain ($p \leq$ 0.044, Fig. 7).

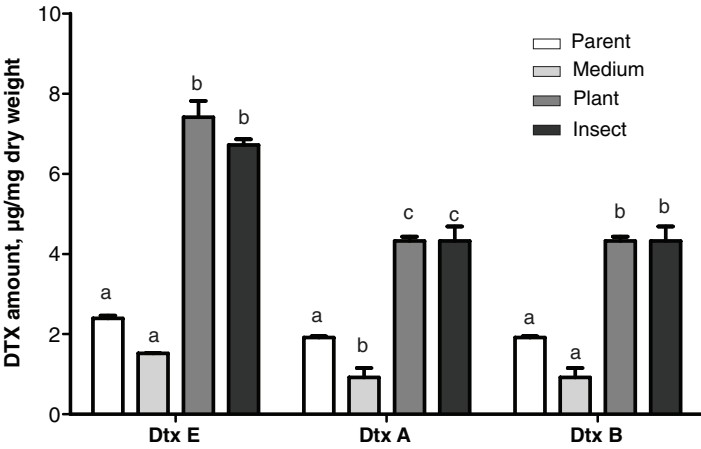

**Figure 5 Amounts of destruxins (Dtx) in Czapek-Dox broth after eight days of cultivation of *Metarhizium robertsii* parent strain MB-1 and its reisolates from the SDAY medium, tomato and wax moth.** Number of samples = 4 per each culture. Different letters indicate significantly different values (Fisher's LSD test, $p < 0.05$).

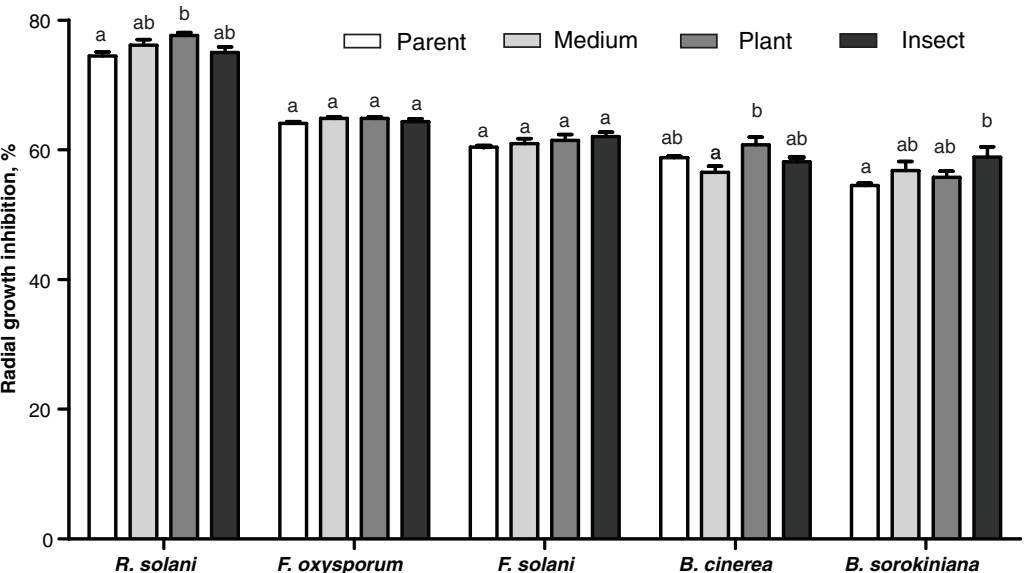

**Figure 6 Antagonistic action of *Metarhizium robertsii* parent strain MB-1 and its reisolates (SDAY medium, tomato and wax moth) against the phytopathogenic fungi.** The fungi assayed were *Rhizoctonia solani*, *Fusarium oxysporum*, *F. solani*, *Botrytis cinerea*, and *Bipolaris sorokiniana*, as inferred from observation of radial growth after 20 days of cocultivation ($n = 3$ for each culture). Different letters indicate significantly different values ($p < 0.05$, Dunn's test).

## DISCUSSION

This article shows that the passaging of *M. robertsii* strain MB-1 for eight generations through an insect or plant caused physiological and biochemical changes associated with suppression of virulence to insects and of lipolytic and proteolytic activities as well as activation of destruxin synthesis. Moreover, there were minor alterations of growth rates (on media containing different carbohydrate sources) and of an antagonistic action on

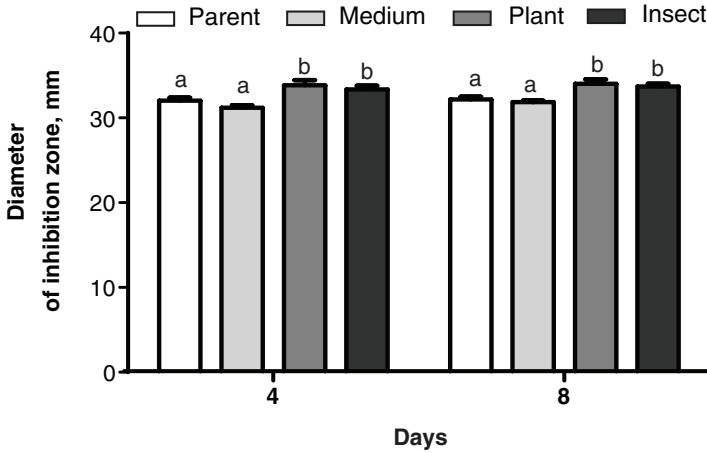

**Figure 7 Growth inhibition of the bacterium *Bacillus pumilus* by *Metarhizium robertsii* parent strain MB-1 and its reisolates (SDAY medium, tomato and wax moth).** Growth inhibition was assayed on days 4 and 8 as the "sterile zone" diameter formed after placing of fungal plugs onto the bacterial culture plated on PDA ($n = 6$). Different letters indicate significantly different values (Fisher's LSD test, $p < 0.05$).

plant-associated fungi and bacteria. The passages of the microorganisms through the live organisms were alternated with plating onto a medium and because the testing of the cultures after the final passage was also performed after one passage on a medium; therefore, we suppose the alterations are explained by epigenetic or selection processes but not by pre-test nutrition of the fungus in question. The constant subculturing on a medium caused only minor changes in comparison to the parent culture, indicating stability of the strain. All the reisolates are characterized by high morphological similarity of the colonies, the absence of sectors, and genetic homogeneity as evidenced by α/β-hydrolase sequencing. This stability points to good prospects of this strain for biological control.

As mentioned above, passaging of entomopathogenic fungi through insects usually affects virulence positively (*Butt et al., 2006*), and in plant-passaged *M. robertsii*, its virulence to insects and the conidiogenesis level also go up (*Hu & Bidochka, 2020*). On the contrary, in the present article, the plant- and insect-passaged reisolates demonstrated significantly lowered virulence to two insect species: (a) the wax moth that was used for the passaging and (b) the Colorado potato beetle, which represents a target host from another insect order. Notably, larvae of these two insects possess strikingly dissimilar epicuticular lipid composition (*Tomilova et al., 2019*; *Kryukov et al., 2022*). In a series of independent studies, it has been reported that passaging through insects may not alter fungal virulence (*Hall, 1980*; *Ignoffo et al., 1982*; *Brownbridge, Costa & Jaronski, 2001*; *Vandenberg & Cantone, 2004*; *Eckard et al., 2014*; *Holderman et al., 2017*; *White et al., 2021*), and only a few articles indicate its weakening (*Hussain et al., 2010*). In the latter work, *M. anisopliae* strain 406 manifested diminution of virulence and major changes in the profile of volatile compounds after one or two passages through the termite *Coptotermes formosanus*. The phenomenon of the virulence decline may be due to the initially high virulence of the examined strain. We have also found previously that *M. robertsii* strain P-72 retains high virulence after prolonged *in vitro* cultivation (over 45 years) though its ability to form

conidia after colonization of insects is lost (*Kryukov et al., 2019*), and its passaging through the wax moth causes a reduction in virulence (V. Kryukov, 2022, unpublished data).

We noticed that the virulence decrease in the reisolates passaged *via* the live organisms correlated with a decline of lipolytic and proteolytic activities. Proteases are important for penetration through the insect cuticle and subsequent growth of the fungus in the host haemocoel. Mutant *M. robertsii* strains with deleted genes of metalloproteases (*Mrmep1* and *Mrmep2*) display a curtailed conidial yield and virulence to the wax moth (*Zhou et al., 2018*). On the other hand, fungal proteases trigger a set of insect immune reactions (*Vilcinskas, 2010*; *Mukherjee & Vilcinskas, 2018*), and overexpression of these enzymes causes strong activation of the phenoloxidase system, thereby possibly resulting in the death of both the pathogen and host (*St Leger et al., 1996b*). The lowering of proteolytic activity may therefore be advantageous from the standpoint of evasion of the host's defense, thus favoring fungal survival and propagation. In terms of plant colonization, proteolytic activity may be a neutral characteristic. Research by *Moonjely et al. (2019)* revealed that a deletion of the subtilisin-like serine protease (Pr1A) gene in *M. robertsii* cuts down its virulence to *Tenebrio molitor* yet does not influence the capacity for rhizoplane and endophytic colonization of barley roots. Their finding is in good agreement with our observed tendency for elevation of tomato colonization frequency in reisolates with weakened protease production. It is also known that endophytic and phytopathogenic fungi possess a significantly smaller set of proteases as compared to *Metarhizium* species (*Hu et al., 2014*). Thus, high levels of proteolytic activity do not seem to be supported by passaging of *Metarhizium* through plants.

The proteases' activity also slightly decreased during subculturing on the medium as compared to the parent culture. Nonetheless, this decrease was not significant and did not affect the virulence of the subcultured reisolate. This phenomenon is often observed when strains of *Metarhizium* are subcultured on media (*e.g.*, *Shah et al., 2007*). Notably, the analysis of changes in the proteolytic activity of reisolates in the present work is of a screening nature and preliminary. We did not analyze the expression of various groups of proteinases, such as subtilisin-like proteases, trypsins, and metalloproteases, although they differ in their impact on the development of pathological processes during a fungal infection (*Vilcinskas, 2010*; *Semenova et al., 2020*). The changes in expression of different proteases during passages through insects and plants should be the focus of future studies.

We detected a sharp increase of production of destruxins in the plant- and insect-passaged reisolates as opposed to a minor decrease after subculturing on a medium, suggesting that these toxins' synthesis is induced by contact with living organisms. Destruxin production negatively correlated with virulence, supporting the idea that these toxins are not major virulence factors. Concentrations of *in vitro*–produced destruxins do not necessarily correlate with insect mortality rates (*Golo et al., 2014*). Mutant strains of *Metarhizium* with zero destruxin production have either minor differences in virulence from the wild-type strain (*Wang et al., 2012*) or no such differences at all (*Donzelli et al., 2012*). *Rios-Moreno et al. (2017)* discovered that *M. brunneum* strains having identical virulence levels against the wax moth produce five- to seven-fold different amounts of destruxin A *in vivo*. Anyway, negative effects of destruxins on cellular and humoral

immunity in insect hosts are well known (*Charnley, 2003*; *Wang et al., 2012*; *Han et al., 2013*). As for plants, *Barelli et al. (2022)* reported that production of destruxins is elevated or their set is wider in various *Metarhizium* species cocultivated on media with beans or maize in comparison to pure fungal cultures; this means that the toxin synthesis can be stimulated by plant root exudates. It is believed that *Metarhizium* destruxins may be an evolutionary relic from ancestral phytopathogenic fungi (*Barelli et al., 2022*). For example, destruxin B has host-selective phytotoxicity and serves as one of major metabolites in *Alternaria* species (*Meena & Samal, 2019*). *Pedras et al. (2001)* noted that *Alternaria brassicae*–resistant *Sinapis alba* metabolizes destruxin B into a less toxic product substantially faster than susceptible species do (*Brassica napus*, *Brassica juncea*, and *Brassica rapa*). This finding suggests that destruxins may facilitate plant colonization by the fungi. This notion is consistent with the results of our study: destruxin production in *M. robertsii* turned out to be enhanced by passaging through the plant. These results, taken together with our data concerning the proteolytic- and lipolytic-activity decline, imply that adaptation to live organisms provokes complex modifications of virulence regulation.

On the basis of these observations, we can hypothesize that high virulence is not necessarily advantageous for this fungus. According to the "trade-off hypothesis" (*Anderson & May, 1982*; *Ewald, 1983*), parasites may evolve toward low but not zero virulence. Their transmission may be interrupted either because of the killing of the host before infection of new individuals or due to insufficient virulence (*Alizon et al., 2009*; *Raymond & Erdos, 2022*). In case of entomopathogenic fungi, the former situation corresponds to host death before complete colonization by a highly virulent fungal strain or poor sporulation on cadavers (*Evison et al., 2015*; *Boucias et al., 2018*), whereas the latter one matches the inability to penetrate the cuticular barrier and a loss of virulence of the fungus. In theory, *in vivo* passaging may eliminate outlying variants (possessing either extremely high or insufficient virulence) and support moderate virulence. In this context, levels of protease and lipase activity do not significantly affect fungi's additional strategy associated with endophytic colonization. Additional experiments are needed to test this supposition.

In our work, aside from similar changes incurred by the passaging through different hosts, some more specific alterations were also detected concerning antimicrobial activity of the fungus and its development on different carbohydrate sources. In particular, the insect-passaged reisolate showed slightly slower growth on the xylose-containing medium, whereas the plant-derived reisolate's growth was retarded by added glucose. Fungal growth dynamics on xylose sources are known to indicate adaptation to development in plants (*Xiao et al., 2012*). *Metarhizium* has a sufficient set of genes related to xylose metabolism (*Duan et al., 2009*). We hypothesize that passaging through insects lowers the synthesis of xylose-metabolizing enzymes.

Although the passaging through the insect or plant did not influence *M. robertsii*'s antagonism toward the intestinal bacteria of the wax moth, the activity was stronger against the plant-associated bacterium *B. pumilus* and phytopathogenic fungi. Notably, it was the passaging through the plant that strengthened the action against *R. solani*. Multiple mechanisms may be involved in this phenomenon. In particular, *M. robertsii* is known to

produce diverse antimicrobial metabolites (helvolic acid, ustilaginoidin, pseurotin, indigotide, and hydroxy-ovalicin) to suppress bacteria (*Sun et al., 2022a*, *2022b*). Moreover, the spectrum of volatile organic compounds emitted by *Metarhizium* fungi inhibits the development of competing bacteria and fungi, including phytopathogenic ones (*Hummadi et al., 2022*). The profile of *Metarhizium* compounds significantly changes during passaging through insects as mentioned above (*Hussain et al., 2010*). Further research will address alterations of the profile of these metabolites under the influence of passaging through plants.

## CONCLUSIONS

This is the first article showing changes in a stable *Metarhizium* strain during passaging through a plant and (separately) through an insect. The study indicates relatively fast physiological alterations of *M. robertsii* during its adaptation to different hosts, thus pointing to ecological plasticity of the fungus. We showed for the first time that the virulence and activity of hydrolytic enzymes (lipases and proteases) can diminish after passaging through a plant or insect, although the production of other virulence factors such as destruxins can greatly increase. The latter is evidence of stimulation of this characteristic by passaging. We suppose that after several iterations of the life cycle through live organisms, reprioritization of virulence factors occurs favoring better adaptation of the fungus to its hosts. The similar nature of virulence factors' changes when the strain passaged through the insect is compared with the strain passaged through the plant may reflect common trends of *M. robertsii* transformation during interactions with different organisms. Nevertheless, some physiological alterations were more specific, including changes of the growth rate on media with different carbohydrate sources and antagonistic properties against phytopathogens. It should be emphasized that the continuous subculturing *in vitro* did not drastically alter virulence of the strain in question, indicating its stability and suitability for practical applications as a microbial control agent against pests and diseases of plants. Subsequent studies on fungal passaging will involve an analysis of epigenetic processes as well as transcriptome and metabolome characterization to elucidate molecular mechanisms of the adaptations of the fungus to different hosts.

## ACKNOWLEDGEMENTS

The authors are indebted to Vladimir Shilo, Olga Polenogova, Evgeniya Buntova, and Tatyana Marchenko (ISEA SB RAS, Novosibirsk) for their help with experiments' organization and technical assistance. The fungal DNA sequencing was performed on the equipment of the core facility Innovative Technologies of Plant Protection at the All-Russian Institute of Plant Protection (St. Petersburg).

### Funding

The research is supported by the Russian Science Foundation (grant # 19-14-00138). The maintenance of laboratory insect cultures was supported by Federal Basic Scientific

Research Program FWGS-2021-0001. The funders had no role in study design, data collection and analysis, decision to publish, or preparation of the manuscript.

## Grant Disclosures

The following grant information was disclosed by the authors:
Russian Science Foundation: 19-14-00138.
Federal Basic Scientific Research Program: FWGS-2021-0001.

## Competing Interests

The authors declare that they have no competing interests.

## Author Contributions

- Oksana G. Tomilova conceived and designed the experiments, performed the experiments, analyzed the data, prepared figures and/or tables, authored or reviewed drafts of the article, and approved the final draft.
- Vadim Y. Kryukov conceived and designed the experiments, analyzed the data, authored or reviewed drafts of the article, and approved the final draft.
- Natalia A. Kryukova performed the experiments, prepared figures and/or tables, authored or reviewed drafts of the article, and approved the final draft.
- Khristina P. Tolokonnikova performed the experiments, authored or reviewed drafts of the article, and approved the final draft.
- Yuri S. Tokarev analyzed the data, prepared figures and/or tables, authored or reviewed drafts of the article, and approved the final draft.
- Arina S. Rumiantseva performed the experiments, authored or reviewed drafts of the article, and approved the final draft.
- Alexander A. Alekseev performed the experiments, authored or reviewed drafts of the article, and approved the final draft.
- Viktor V. Glupov conceived and designed the experiments, authored or reviewed drafts of the article, and approved the final draft.

## DNA Deposition

The following information was supplied regarding the deposition of DNA sequences:
The sequences of alpha/beta-hydrolase gene fragment are available in the GenBank database: OP985335–OP985338.

## Data Availability

The raw data on the plant colonization; growth on different media; mortality of *Galleria mellonella* and *Leptinotarsa decemlineata*; enzyme activities, destruxin production, and antagonism against fungi and bacteria are available in the Supplemental File.

## Supplemental Information

Supplemental information for this article can be found online at http://dx.doi.org/10.7717/peerj.15726#supplemental-information.

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
