# Peer review of "Effects of passages through an insect or a plant on virulence and physiological properties of the fungus Metarhizium robertsii"

_PeerJ, doi:10.7717/peerj.15726_

## Round 0.1 · original submission · Major Revisions

It sounds like an interesting and important study that sheds light on the multifunctional lifestyle of Metarhizium robertsii and how it adapts to different environments. The finding that passaging the fungus through different hosts and media results in physiological alterations and changes in virulence factors is particularly intriguing and suggests that the fungus undergoes a reprioritization of its metabolic pathways depending on its environment. I agree that your paper should be of interest to a broad audience, including researchers studying fungal development and evolution, mycotoxins, and biological control of pests and diseases of plants. Authors are requested to revise the manuscript according the reviewers comments.

Reviewer 1 ·

Basic reporting

.

Experimental design

.

Validity of the findings

.

Additional comments

This manuscript presents valuable information on how Metarhizium robertsii's virulence and physiological properties are affected through passages on insects, plants, and media. This is an important contribution to biological control programs. The manuscript is well written and the data presented support the conclusions. A few corrections were done in the manuscript and few recommendations are included to improve it.

Annotated reviews are not available for download in order to protect the identity of reviewers who chose to remain anonymous.

Reviewer 2 ·

Basic reporting

It is an interesting study of altered pathogenic behaviours which provide insights into the physiological alterations of Metarhizium robertsii MB-1 after passaging through different organisms and media. The findings could be useful for understanding the adaptability of entomopathogenic fungi and their potential application in controlling insect pests and plant diseases.

The abstract and introduction do not provide a clear context for the study, which makes it difficult to understand the significance of the findings. For example, it is unclear why it is important to compare the physiological alterations of the reisolates after passaging through insect or plant organisms.

The introduction is lacking the latest citations.

Experimental design

Please provide the accession number or other identification details of studies cultures Metarhizium robertsii MB-1 and G. mellonella.

How the methodology of passaging through plants, insects and media was standardized? please provide suitable references.

Line 186-187 suspension (107 conidia/mL) for 15 s? or 10*8 conidia/mL

Validity of the findings

Findings are appropriate.

It is suggested to include pictures of inhibitory assays.

Discussion lines 413-426 are repetitive of results.

Provide the latest citations in the discussion.

Reviewer 3 ·

Basic reporting

Clear, unambiguous, professional English language used throughout. Introduction & background to show context. Literature well referenced & relevant. The structure conforms to PeerJ standards, discipline norm. Figures are relevant, high quality, well labelled & described and properly cited in the text. Raw data supplied. However, a few corrections in references style are to be done during typesetting in line number 582, 602, 606, 609, 611, 618, 629, 633, 644, 650, 660, 678, 681, 683, and so on (journal volume number in brackets should not be in bold).

Experimental design

Original primary research within Scope of the journal. Research question well defined, relevant & meaningful. It is stated how the research fills an identified knowledge gap. Rigorous investigation performed to a high technical & ethical standard. Methods described with sufficient detail & information to replicate.

Validity of the findings

Meaningful replication is encouraged where rationale & benefit to literature is clearly stated. All underlying data have been provided; they are robust, statistically sound, & controlled. The present finding were thoroughly discussed and compared with the findings of previous workers. The authors clearly mentioned that the analysis of changes in the proteolytic activity of reisolates in the present work is screening in nature and preliminary. Conclusions are well stated, linked to original research question & limited to supporting results.

Additional comments

The article presents several important findings. Like destruxin production do not correlate with insect mortality (line 478), destruxin may facilitate plant colonization and inactivation of destruxin may inactivate fungal invaders. In particular, passaging of a stable strain through an insect and a plant caused physiological alterations, including the overproduction of major secondary metabolites called destruxins, a decrease in protease and lipase activities, and a lowering of virulence toward insects as compared to the parent strain and a reisolate subcultured on a medium. We believe that reprioritization of the fungal virulence factors occurs during the adaptation of M. robertsii to insects, plants, and media. Based on the observations, the authors hypothesized that high virulence is not necessarily advantageous for the fungus. The article also indicated the area of focus for future studies. The changes in the expression of different proteases during passages through insects and plants should be the focus of future studies. The article is of interest to a broad audience, including researchers studying fungal development and evolution, mycotoxins, and biological control of pests and diseases of plants.

---

## Round 0.2 · accepted · Accept

The authors have addressed all of the reviewers' comments.

Reviewer 2 ·

Basic reporting

The manuscript is now acceptable.

Experimental design

Very well improved.

Validity of the findings

Satisfactory.

Additional comments

NA